# The (Non) Economic Properties of the Law

Leo Katz [1,*] and Alvaro Sandroni [2]

1   University of Pennsylvania Law School, 3501 Sansom Street, Philadelphia, PA 19104, USA
2   Department of Managerial Economics and Decision Sciences, Kellogg School of Management,
    Northwestern University, Evanston, IL 60208, USA; sandroni@kellogg.northwestern.edu
*   Correspondence: lkatz@law.upenn.edu

**Abstract:** This paper shows that the logical properties of constraints imposed by law are fundamentally different from other constraints considered in economics such as budget constraints and bounded rationality constraints, such as the ones based on inattention or shortlisting. This suggests that to fully incorporate law into economics may require a revision of economic theory.

**Keywords:** law; order; rationalizability

## 1. Introduction

A workhorse of economics is the traditional model of full rationality, where the decision maker maximizes a utility function $u$ subject to a feasibility constraint $B$. As is well known (Samuelson (1938)) [1], the empirical content of this model is given by the weak axiom of revealed choice (WARP). However, psychologists and economists have uncovered many types of anomalous behavior that violate WARP. Motivated by these observations, models of bounded rationality were developed. In an important branch of this literature, the decision maker maximizes an utility function $u$ subject to a consideration set $\psi(B)$ that belongs to $B$. For example, Masatlioglu, Nakajima and Ozbay (2012) [2] study models where the decision maker can be inattentive, and only pays attention to a subset $\psi(B) \subseteq B$ of options.

Models of consideration sets $\psi(B)$ involve two types of choice functions (to ease the terminology we will not distinguish choice correspondences and choice functions. We refer to a choice function as any function that takes a set $B$ as input and returns a subset of $B$ as output). The first choice function is the consideration function $\psi$ that takes a feasible set $B$, as input, and returns a consideration set $\psi(B) \subseteq B$ as output. The second choice function is the decision function $D$ that also takes a feasible set $B$, as input, and returns a decision set $D(B) \subseteq B$ as output, where $D(B) = \text{argmax } u(x)$ subject to $x \in \psi(B)$.

In this paper, we deal with one specific type of consideration function: the law. That is, given a feasibility set $B$, the consideration set $L(B) \subseteq B$ determines what is both feasible and legal. Our main objective in this paper is to show a difficulty. It is hard to find out what are, if any, the general properties on the consideration function given by the law. To this end, we show three examples. The first example shows that the law violates WARP. Thus, this example casts doubt on the often repeated idea that the law maximizes a social welfare function. The second example shows a deeper violation of WARP: the law does not maximize an asymmetric binary relation either. Hence, the law violates the seemingly natural principle that "illegal options are the options deemed inferior to other feasible options," if by inferior we mean that for some asymmetric binary relation $\succ$, $x \succ y$ whenever an option $y$ is deemed inferior to an alternative option $x$.

The logical structure of our two initial examples of WARP violations may seem familiar to many readers because they are the type of WARP violations coming from psychology (and experimental economics) that motivated the development of the literature of bounded rationality in economics. There is, however, a fundamental conceptual difference between

our examples and the motivating examples that came from psychology. The examples that came from psychology show WARP violations on the decision function $D$. We show examples of WARP violations on the consideration function $L$. This is a fundamental difference: To understand this difference, consider the bounded rationality models that were developed to accommodate the examples coming from psychology (e.g., the short-listing theory of Manzini and Mariotti (2007) [3] and the inattention theory of Lleras et al. (2017)) [4]. None of them can accommodate our examples of WARP violations, but both of them can accommodate seemingly analogous examples of WARP violations that come from psychology.

Take, for example, the inattention model of Lleras et al. (2017) [4]. This model is based on consideration sets $\psi$ called consideration filters. In our initial examples, we show that the law $L$ is such that the decisions of a law abiding citizen can be cyclic. The inattention model of Lleras et al. (2017) [4] can accommodate cyclic choices. However, the law $L$ is a consideration set. It is not the decision function of the law abiding citizen. More to the point, the law violates the properties of consideration filters in Lleras et al. (2017) [4].

The law also violates the properties of other consideration sets such as the ones in the shortlisting theories of Manzini and Mariotti (2007) [3]. By contrast, our first two examples do in fact satisfy the attention filter type of consideration function taken up in the inattention theory of Masatlioglu, Nakajima and Ozbay (2012) [2]. In our third example, however, we show that the law is not an attention filter either. All of this suggests that existing economic models cannot accommodate the law, and that to properly incorporate the law into economics will require significant re-thinking of economic principles. That is, just as behavior observed by the psychologists and economists challenged the traditional model of full rationality, the law provides a new challenge to modern economic theory that has not yet been properly addressed.

Our examples also motivate a fundamental question for which we have no answer: What are the abstract properties of the law? Furthermore, perhaps even more fundamentally, does the law have any such abstract properties? Can we simply consider the law as an arbitrary set of exogenously given constraints imposed on the decision-maker?

Just to be a bit clearer on what we mean by abstract properties, consider a budget constraint. It represents what the decision-maker can afford (i.e., what the decision-maker can purchase and, hence, what the decision-maker can obtain without the use of illegal methods such as stealing). Now a budget contraint does have some abstract properties. For example, if a new good is introduced in the market then, all else equal, what the decision-maker could afford previously the decision-maker can still afford after the good is introduced. This follows because if the decision-maker does not buy any of the newly introduced goods, then the decision-maker has not spent any of his wealth on these goods. Provided that prices and their wealth remain the same, he can afford any bundle of goods that he could afford before. This is a typical example of a principle that underlies, only in part naturally, the theory of the consumer. What we are looking for are equally simple and obvious abstract principles that could underlie a broader (axiomatized) theory of legal decision-making.

Consider the decision function $D$ of a law abiding citizen (i.e., $D(B) = \text{argmax } u(x)$ subject to $x \in L(B)$). So, this is the decision function of an agent who abides by the law $L$ and subject to that constraint, maximizes an utility function. In order to axiomatize the choices of a law abiding citizen, it is necessary to determine what are the properties, if any, that one can assume for the law $L$. A stumbling block in this endeavor is that no matter how simple and intuitive is the abstract principle that we have considered, we could always show, at least so far, that the law violates it. (When we speak of "the law," we are referring to a set of widely shared basic legal principles (the ones involved in our examples) that are rooted in equally widely shared principles of commonsense morality.) Consider the idea that if the law deems an option to be illegal for a decision-maker, then there must be something else that the decision-maker could have done that is deemed to be better (in the traditional sense of better and worse given by an asymmetric binary relation $\succ$). This

may seem a natural candidate for an abstract principle of the law. Yet, as we show, the law violates it.

Now clearly, we cannot show a negative. That is, we cannot show that the law violates any abstract principle. Nor do we believe in such conclusion. It seems (to us) implausible that the law does not satisfy any conceivable abstract principle. However, as our examples show, it is not an easy task to determine what these principles might be.

This paper is organized as follows: In Section 2, we show that the law does not maximize a utility function and does not maximize an asymmetric binary relation either. In Section 3, we show that the law does not satisfy the properties of the consideration sets in models of bounded rationality theories in economics. Thus, as far as we know, there exists no model in economics that can accommodate the law. In Section 4, we discuss the significance of these examples to the development of an axiomatized theory of legal decision making.

## 2. The Law

Let $A$ be a finite set of alternatives. An issue $B$ is a non-empty subset of $A$. Let $\mathcal{B}$ be the set of all issues. A legal system is a mapping $L$ that takes an issue $B$, as input, and returns, as output, a non-empty subset, $L(B)$, of $B$. The set $L(B)$ consists of all the legal options when $B$ is the feasible set of alternatives (there are, of course, greater and lesser crimes. The legal/illegal dichotomy just simplifies the language). Thus, a legal system is a mapping $L : \mathcal{B} \longrightarrow \mathcal{B}$ such that $\varnothing \neq L(B) \subseteq B$. We may refer to $L$ as the law.

Some readers may be concerned with the idea of the law in the singular. After all, the law can differ in different times and places. However, our examples are based on widely shared legal principles. After each example, we discuss the generality of these basic principles.

Let $\succeq$ be a binary relation on $A$. Let $\succ$ be the asymmetric binary relation such that $x \succ y$ if $x \succeq y$ and $y \not\succeq x$ (i.e., it is not the case that $y \succeq x$). An order $\succeq$ is a complete and transitive binary relation.

**Definition 1.** *A legal system L is* **ordered** *if there is an order $\succeq$ such that for any issue B and option $x \in B$,*

$$x \in L(B) \Leftrightarrow x \succeq y \text{ for all } y \in B.$$

In an ordered law, all options are ranked and the legal ones are the highest-ranking ones. An option outranked by another is illegal. The highest-ranking choices are not necessarily unique. If two alternatives have the same (top-)rank, then they are both legal.

**Definition 2.** *A legal system L is* **rationalizable** *if there is an asymmetric binary relation $\succ$ such that for every issue B, and for every $y \in B$,*

$$y \notin L(B) \Leftrightarrow x \succ y \text{ for some } x \in B.$$

The binary relation $\succ$ need not be complete or transitive. So, to be rationalizable is weaker than to be ordered. The binary relation $\succ$ also need not be an individual or social preference. That is, $x \succ y$ simply states that $x$ is strictly better than $y$, according to $\succ$. Thus, rationalizable legal systems are the ones such that "illegal options are the ones strictly inferior to some other feasible option," where by strictly inferior we mean under some asymmetric binary relation $\succ$. In abstract, this may seem an unassailable principle and yet the law violates it.

**Example 1.** *The law is not ordered.*

A decision maker is under deadly attack. His options are: $(x)$ to kill his attacker; $(y)$ to stand their ground and seriously risk being killed or injured; or $(z)$ to escape. Under these circumstances, the first option is illegal and the latter two are legal. However, if

their options were just to kill or seriously risk being killed then killing their attacker is self-defense and, hence, legal. The law is

$$L(x,y) = (x,y); L(y,z) = (y,z); L(x,z) = (z); L(x,y,z) = (y,z). \tag{1}$$

A direct implication of example (1) is that the law does not maximize a utility function. This challenges the idea that the law has a greater goal, such as maximization of a social welfare function.

Laws with a logical structure as in (1) can produce choices that are legal, fully rational, and cyclic (see Katz and Sandroni (2017)) [5]. Consider a law abiding citizen who prefers $x$ to $y$ to $z$ and has a high disutility for breaking the law (either because of fear of punishment or because of a moral constraint) and so only takes legal options. Then, the optimal binary choices are $x$ over $y$, $y$ over $z$ and $z$ over $x$. These choices are cyclic and stable in the sense that the decision maker may not reconsider them upon careful consideration. They optimize overall preferences based on normatively compelling principles (optimization and respect for the law) and not on cognitive limitation. Their root cause is the logical structure of the law.

**The legal principles in Example 1**

In Example 1, $L(x,y) = (x,y); L(y,z) = (y,z); L(x,z) = (z)$ suffices to show that the law is not ordered. $L(x,y) = (x,y)$ is based on an elementary idea of self-defense. If the only option is to seriously risk death, then to kill their attacker $(x)$ is legal in basically any legal system that allows for self-defense. Option $(y)$ is also legal because otherwise the law would require harming someone else, which only occurs under very special circumstances. As for $L(y,z) = (y,z)$ this simply reflects respect for the decision maker's autonomy so long as it is only their own life that he is risking. Finally, $L(x,z) = (z)$ is also quite universal because if the options are merely to kill the attacker $(x)$ or to escape and not to kill the attacker $(z)$, then the attacker poses no real threat whatsoever to the decision-maker (i.e., $y$ is not available to the decision-maker). In this case, $z$ is typically the only legal option.

As mentioned, once $L(x,y) = (x,y); L(y,z) = (y,z); L(x,z) = (z)$, the law is not ordered regardless of what is legal when all three options are available. We chose $L(x,y,z) = (y,z)$ because if the decision-maker can escape and avoid harm, then the decision-maker is not allowed to kill legally (i.e., $x \notin L(x,y,z)$) in most legal systems. However, this principle is not quite universal. For example, in some states in the US with "stand your ground" laws it is permitted to kill an attacker legally even when the decision-maker could avoid harm by escaping. However, it is generally conceded that the moral intuition that killing should not be permitted when escape is possible has a great deal of force.

**Example 2.** *The law is not rationalizable.*

Some valuable property, e.g., a wallet, of the decision maker can be taken away. The loss of the wallet can only be stopped by killing the robber. So, the options are: $(x)$ to kill the robber, and $(y)$ to give up the wallet. In this case, $y$ is the only legal option. Now consider a third option. The loss of the wallet can be prevented without killing (say by placing it out of reach), but this results in great harm to the decision maker (say a serious beating). So, now there is a third option $(z)$ to keep the wallet and endure great harm. Even though option $z$ can be avoided, if the decision maker gives up their property (i.e., chooses $y$), the law now considers all three options $x$, $y$ and $z$ legal. The law is

$$L(x,y) = (y); L(y,z) = (y,z); L(x,z) = (x,z); L(x,y,z) = (x,y,z). \tag{2}$$

This law is not rationalizable because $L(x,y) = y$ implies $y \succ x$ which is contradicted by $x \in L(x,y,z)$. The law in (2) can also produce cyclic choices when the law abiding citizen prefers $x$ to $z$ to $y$.

**The legal principles in Example 2**

People are often puzzled when they first encounter Example 2. A common reaction is to ask how it is possible to kill legally when killing can be prevented by giving up property. To be sure, the first part of this example, $L(x, y) = (y)$ is generally seen as uncontroversial. If the option is to kill the robber or to give up some property (and there is no possibility of harm to the decision-maker), then killing is not generally not allowed and the only legal option is to give up the wallet. On the other hand, people are often puzzled by the idea that killing the robber $x$ is legal ($x \in L(x, y, z)$), when the decision-maker can choose $z$ and therefore not kill the robber and not suffer any physical harm either (Model Penal Code, 3.04 and 3.06). Perhaps surprisingly, this aspect of the law applies far more generally than in states with "stand your ground" laws.

Example (2) can be understood using common sense and analogous properties can be found in every major branch of the law including property, contracts, torts and criminal law. What lurks behind the law is the need to balance conflicting principles. One principle is that " people should not have to surrender their property merely because others want it." The other is that "life is more important than property." In the first case, when the only choice is between the decision maker's property and the life of the assailant, property loses out and, hence, deadly force is not allowed (Hence, $L(x, y) = (y)$). However, once other options are injected in, such as being severely beaten, then the idea that people should not need to acquiesce to robbery gains greater force, even if this is an option that can be rejected. Thus, in a choice between $x$, $y$, and $z$, the decision-maker can legally refuse to surrender their property (i.e., not choose $y$), but if so, then the robber is going to beat them up, i.e., $z$ occurs, unless the decision maker instead chooses $x$ and stops the robber by using deadly force. However, this time it is often legal to choose $x$.

## 3. Relationship to the Bounded Rationality Literature

Let us now revisit the consideration sets $\psi(B) \subseteq B$ of three important theories: the shortlisting theory of Manzini and Mariotti (2007) [3], the inattention theory of Lleras et al. (2017) [4] and the inattention theory of Masatlioglu, Nakajima and Ozbay (2012) [2].

The consideration function $\psi : \mathcal{B} \longrightarrow \mathcal{B}$ in the shortlisting theory of Manzini and Mariotti (2007) [3] selects from the option set those for which, for some asymmetric binary relation $\succ_1$

$$\psi(B) = \{x \in B \mid \nexists \ y \in B \text{ for which } y \succ_1 x\}. \tag{3}$$

The consideration function in the inattention theory of Lleras et al. (2017) [4], called consideration filters, and the psychological constraints in Cherepanov et al. (2013) [6] is such that

$$\text{if } B \subseteq B^* \text{ then } \psi(B^*) \bigcap B \subseteq \psi(B). \tag{4}$$

The consideration function in the inattention theory of Masatlioglu, Nakajima and Ozbay (2012) [2], called attention filters, is such that

$$\psi(B) = \psi(B \setminus y) \text{ whenever } y \notin \psi(B). \tag{5}$$

**Remark 1.** *The law L in (2) is not a consideration filter and it is not a consideration set in shortlisting theory either.*

We expressed this point as a remark (as opposed to a new example) because **our** example (2) suffices to show it. A simple argument that shows that the law $L$ in (2) violates (3) and (4) is as follows: The law violates (4) because if $B^* = (x, y, z)$ and $B = (x, y)$ then $L(B^*) \bigcap B = (x, y) \nsubseteq (y) = L(x, y)$. It violates (3) because if not then $(x, y) \subseteq L(x, y, z) \Longrightarrow y \nsucc_1 x \Longrightarrow x \in L(x, y)$ and this is contradicted by $L(x, y) = (y)$.

The law $L$ in (2) does not violate (5) and, hence, is an attention filter. This follows because in (2), the only issue in which there is an option $x \notin L(B)$ is in the binary issue

$(x, y)$. An attention filter is neither stronger nor weaker condition than rationalizability. Hence, it needs to be dealt with separately.

**Example 3.** *The law is not an attention filter.*

An emergency room has three patients: Al $(x)$, Bea $(y)$ and Chloe $(z)$. Al has the most serious injuries (e.g., two legs hurt), followed by Bea (e.g., one leg hurt) and then Chloe (one finger hurt). Legality here means the patient that the doctor is legally allowed to treat first. Typically, the patient with the most serious injuries has priority, but let us say that Al would like to "alienate" their treatment priority to their wife Chloe. In the presence of Bea he generally will not be allowed to. So, it is legal for the doctor to treat Al, but treating either Bea or Chloe is illegal. However, if Bea is not part of the picture then the doctor would be allowed to heed Al's request to treat Chloe first. In this case, the law is

$$L(x, y) = x; \; L(y, z) = y; \; L(x, z) = (x, z); \; L(x, y, z) = x. \tag{6}$$

This law is rationalizable (by $x \succ y$ and $y \succ z$), but it does not satisfy (5). If $B = (x, y, z)$ then $y \notin L(B)$ and $L(x, y, z) \neq L(x, z)$. So, this law is not an attention filter.

**The legal principles in Example 3**

Example 3 is based on two principles. The first one is that there is some priority rule in medical care that doctors must respect. In Example 3, this priority rule is based on the gravity of the injuries, but this is clearly incidental to the example. As long as there is some priority rule in medical care and Al has priority over Bea under that rule, then $y \notin L(x, y, z)$. While quite general, this principle is not universal because in some cases, doctors have complete discretion over who to give priority treatment. So, here we are limiting ourselves to the case in which there is some priority rule in medical care. However, once there is such rule, then our next principle is really just the Pareto principle: Al and Chloe can switch places so long as Bea is not adversely affected.

## 4. What Are the Abstract Properties of the Law?

Examples (1), (2), (6) show legal systems with properties that depart from constraints commonly seen in economics. In this section, we discuss the significance of these examples for the development of an axiomatized theory of legal decision making.

An utility function $u$ is a mapping $u : A \longrightarrow \Re$, where $\Re$ is the set of real numbers. An utility function $u$ is without indifferences if $u(x) \neq u(y)$ for all $x \in A$ and $y \in A$, $x \neq y$.

**Definition 3.** *Given a law $L$, a choice function $C$ is **legal** if $C(B) \in L(B)$ for any $B \in \mathcal{B}$.*

That is, a choice function $C$ is legal if all choices that $C$ produces are legal.

**Definition 4.** *Given a law $L$, $C$ is a **law-abiding citizen choice function** if $C$ is legal and there exists an utility function $u$ without indifferences such that for any $B \in \mathcal{B}$,*

$$u(C(B)) > u(y) \text{ for all } y \in L(B), y \neq C(B).$$

A law-abiding citizen choice function is such that the choice $C(B)$ is the best alternative among the legal options, for some utility function $u$.

Our examples might suggest replacing the set of feasible options $B$ with the set $L(B)$ of legal options, with no restrictions on the legal constraint function $L$. However, this would merely be an study on arbitrary consideration sets $L$. The arbitrariness of $L$ would not make the axiomatization of law-abiding citizen choice functions technically difficult. In fact, that exercise would follow directly from known results that can be found in Richter (1966) [7]. We show these results here only to emphasize that the main difficulty in the development of a proper theory of law abiding choices is in the determination of what

are the general properties of the law. We also refer the reader to Richter and Rubinstein (2020) [8] for a comparison between social norms and general equilibrium models.

Given a law $L$, and a choice function $C$, let $R$ $(= R^{C,L})$ be a binary relation such that given any two options $x \in A$ and $y \in A$, $x \neq y$,

$$x \ R \ y \ \text{ if and only if } \{x, y\} \subseteq L(B) \text{ and } x = C(B) \text{ for some } B \in \mathcal{B}.$$

So, $x \ R \ y$ indicates that $x$ is revealed to be preferred to $y$.

**Proposition 1.** *Given a law $L$, $C$ is a law-abiding citizen choice function if and only if $C$ is legal and $R$ is an asymmetric and acyclical binary relation.*

**Proof.** Let $C$ be a legal choice function. Assume that $R$ is an asymmetric and acyclical binary relation. By topological ordering, $R$ may be extended to an asymmetric order $P$ (see Cormen et al. (2001)) [9]. Let $u$ be the associated (with $P$) utility function without indifferences. Consider any issue $B \in \mathcal{B}$. If $y \in L(B)$ and $C(B) \in L(B)$, $y \neq C(B)$, then $C(B) \ R \ y \implies C(B) \ P \ y$. Thus, $C$ is a law-abiding citizen choice function.

Assume that $C$ is a law-abiding citizen choice function. Let $P$ be the asymmetric preference order associated with the utility function $u$. Assume, by contradiction, that $x \ R \ y$ and $y \ R \ x$, $x \neq y$. Then, for some $B \in \mathcal{B}$, $\{x, y\} \subseteq L(B)$ and $x = C(B)$ and for some $B' \in \mathcal{B}$, $\{x, y\} \subseteq L(B')$ and $y = C(B')$. So, $x \ P \ y$ and $y \ P \ x$. This contradicts the asymmetry of $P$. Furthermore, assume, by contradiction, that $x \ R \ y$ and $y \ R \ z$, and $z \ R \ x$, $x \neq y \neq z$. Then, for some $B \in \mathcal{B}$, $\{x, y\} \subseteq L(B)$ and $x = C(B)$; for some $B' \in \mathcal{B}$, $\{y, z\} \subseteq L(B')$ and $y = C(B')$; for some $B'' \in \mathcal{B}$, $\{x, z\} \subseteq L(B'')$ and $z = C(B'')$. Thus, $x \ P \ y$, $y \ P \ z$, and $z \ P \ x$. This contradicts the transitivity of $P$.

Proposition 1 shows what an axiomatized theory of law-abiding choices looks like when the consideration set of legal options is arbitrary and is not assumed to have any underlying abstract properties. As it is easy to see, this is not an interesting theory of law-abiding choices in the same way that it would not be an interesting choice of limited attention if we assume limited attention to be just any exogenously given constraint. The development of a proper theory of law abiding choices required the determination of some general properties of the law, if they exist. However, as our examples show, it is not an easy task to determine what these properties might be. □

## 5. Conclusions

Phenomena studied by psychologists and experimental economists provided the original impulse for the development of behavioral economics. The law delivers a similar impulse, but the law is fundamentally different from the phenomena originally uncovered by psychologists and experimental economists. In addition, logical properties of legal constraints differ from other constraints considered in economics such as inattention or shortlisting. General economic principles studied so far are likely to run afoul of how the law actually operates. This applies to seemingly unassailable principles such as "illegal options are the ones inferior to some other feasible option." This poses a new challenge on how to properly incorporate law into economics.

**Author Contributions:** Both authors contributed to this paper in the same way. Both authors have read and agreed to the published version of the manuscript.

**Funding:** This research received no external funding.

**Institutional Review Board Statement:** Do not apply.

**Informed Consent Statement:** Do not apply.

**Data Availability Statement:** Do not apply.

**Acknowledgments:** None.

**Conflicts of Interest:** The authors declare no conflict of interest.

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
