# Peer review of "The (Non) Economic Properties of the Law"

_games, doi:10.3390/g12010026_

Round 1

Reviewer 1 Report

This is a short paper – it shows how “weak” can “Law” be!! I find it interesting however I was expecting more in terms of content and results.

The definition of legal options (just as a subset of set of available alternatives) is acceptable however makes it fairly difficult to rationalise as this paper shows.

The examples are well chosen – the illustrations are all correct.

There is not positive result here. I wish the author has worked more on this concept. Incidentally, I am sure the author is aware of all the recent papers by Ariel Rubinstein.

There are a few typos (such as, in Definition 3, “there”). I would strongly suggest that the author(s) take(s) a very careful look and do(es) a sound editorial check.

Author Response

Dear Referee 1:

We are very pleased that you found our paper to be interesting, and that you found our examples to be well chosen, and our illustrations to be correct. However, you also expected more in terms of content and results. Following your suggestion (which is quite consistent with the letter of the editor and referee 2's report) we added a new section in this revision. This is Section 5 entitled "What are the Abstract Properties of the Law?" In this section, we make clear that we are not answering a question in this paper. Rather we asking a new question and showing why this question is so hard to answer. The question is of course: What are the Abstract Properties of the Law?" We believe that, like budget constraints and bounded rationality constraints, the law should have general abstract properties as well. It would, to our mind, be extremely surprising (and almost inconceivable to legal scholars who are often preoccupied with internal consistency of the law) that the law, unlike any other constraint in economics, would have no general abstract properties. But if the law does have general properties, then what are they? Our examples show that these general properties are not easy to pin down. Naturally, one may think that the difficulty is in our abstract set-up of thinking about the law as a consideration set. It would, of course, be an interesting conclusion if it turns out that one cannot think about the law in these terms, especially because we can (and we do) think about other constraints in economics as consideration sets and also because in all of the examples we can think of, some of which are described in the paper, this is a perfectly suitable way of describing the law. But all of these are, of course, questions to be addressed in future research. In the new section 5 we point out that the difficulty in determining what are the general properties of the law is a chief obstacle in the development of an axiomatized theory of legal decisions.

We also expanded the introduction to better motivate our examples early on in the paper. We now mention that our examples show that there are no economic theories (that we know of) that can accommodate the law and that our examples also show that the law does not maximize a utility function or even an asymmetric binary relation. This point is significant because the idea that the law has an ulterior goal is quite prevalent among legal scholars.
In your report, you also mentioned recent papers by Ariel Rubinstein. This was a serious oversight in the previous version of the paper. In this revision, we cite the recent paper by Richter and Rubinstein "The Permissible and the Forbidden," JET, 2020. Like our comparison between the law and economic analysis, Richter and Rubinstein also make a comparison between social norms and general equilibrium models. Thus in the new section 5, page 11, first paragraph we say that "We also refer the reader to Richter and Rubinstein (2020) for a comparison between social norms and general equilibrium models."

You also mentioned a typo in Definition 3, "there." We fixed this typo in this revision and also carefully looked for other errors and typos through the manuscript.

Thank you for the constructive comments that you made on our work. Your comments proved very helpful and we think this revision is much improved because of them. We hope that we revised the paper along the lines that you expected and that we took full advantage of your report.

Reviewer 2 Report

The authors shows, through a set of counterexamples, that the existing models of bounded rationality cannot be satisfactorily used to accommodate the law. They conclude that the law is not ordered, it is not rationalizable, it is not a consideration filter, it is not a consideration set, it is not an attention filter, and it does not satisfy the indifference axiom.

I am a total outsider to the literature on law and economics. My expertise is in experimental and behavioral economics. I find the topic and the underlying general research question interesting. I am puzzled, however, why the listed properties are relevant at all. In other words, why would anyone expect the law to satisfy any of them. Why should the law be ordered, or rationalizable, etc? Why can't we consider the law as a set of exogenously given constraints imposed on the decision-maker?

Also, the law does not strike me as anything similar to a hard budget constraint (or a hard feasibility constraint). The decision-maker, after careful consideration or without it, might decide to violate the law.

Even if the editor considers that the contributions of this paper warrant publication, the authors should provide a more careful and deeper discussion on the motivation of the study. Similarly, I do not see how one can talk about "the law" in general. The authors must be much more specific than that.

A short list of typos.
(line 1, page 6) Remove the dot after "et" and the comma after "al." in the reference.
(line 9, page 6) Is the statement in boldface an other example? Why does it have a proof while the others do not.
(definition 3, page 7) Write "three" instead of "thre".
(line 4, Section 5, page 8) Write "different" instead of "differ". Or remove "are".
(line 1, Appendix, page 8) Is the statement in boldface an other example? Why does it have a proof while the others do not.

One more comment about presentation.
Why do you need such a short appendix? Why can't that text be moved to the main part?

Author Response

Dear Referee 2:

We are happy to read that you found "the topic and the underlying general research question interesting." However, in your report, you expressed reservations as to the relevance of the properties that we addressed. In your report, you wrote: "In other words, why would anyone expect the law to satisfy any of them. Why should the law be ordered, or rationalizable, etc? Why can't we consider the law as a set of exogenously given constraints imposed on the decision-maker? Also, the law does not strike me as anything similar to a hard budget constraint (or a hard feasibility constraint). The decision-maker, after careful consideration or without it, might decide to violate the law."
Following your question, we revised the paper by adding in the introduction more explicit motivation for our examples (which is also consistent with your comment that "the authors should provide a more careful and deeper discussion on the motivation of the study"), and we also added a new section (section 5 in this revision) that explicitly addresses your question of "Why can't we consider the law as a set of exogenously given constraints imposed on the decision-maker?" Finally, we note that the principles of law that generate these odd violations are shared by all legal systems as well as the commonsense morality on which they are all based. They are not an artifact of something some particular and eccentric legislator decided to do.

Before getting to our changes in the paper, we would like to informally discuss some matters with you. There is no expectation that the law would or should have the same properties as other constraints in economics such as budget constraints or bounded rationality constraints. However, it would, to our mind, be extremely surprising (and almost inconceivable to legal scholars who are fundamentally interested in internal consistency of the law and also on the overall purpose of the law) if the law, unlike any other constraint in economics, turned out to have no general abstract properties at all. But if the law does have some general properties, then what are they? Our examples show that these general properties are not easy to pin down and we are fairly sure that no one really knows what these properties might be. Even an extremely natural candidate property for the law such "illegal options are those options deemed inferior by some binary relation" is violated by the law. Thus, in this revision, we make clear that we are not answering a question in this paper. Rather we asking a new question and showing why this question is so hard to answer. The question is: What are the Abstract Properties of the Law? In addition, we are also showing that no economic model that we know of can accommodate the law and that it is also quite difficult to determine if the law maximizes anything or has some overall objective such as maximization of social welfare.

Naturally, one may think that the difficulty is in our abstract set-up of thinking about the law as a consideration set. Perhaps this is what underlies your comment "Similarly, I do not see how one can talk about "the law" in general." It would be an interesting conclusion to our questions if it turns out that one cannot think about the law as we describe (i.e., as a consideration set), especially because we can (and we do) think about many other constraints in economics as a consideration sets and, more importantly, also because in all of the examples we can think of, some of which are described in the paper, this is a perfectly suitable way of describing the law. But all of this is to be addressed in future research.
Now let us address the explicit changes in the paper made in response to your comment. First, we added the following paragraphs in the introduction.
"Our examples also motivate a fundamental question for which we have no answer: What are the abstract properties of the law ? And perhaps even more fundamentally, does the law have any such abstract properties? Can we simply consider the law as an arbitrary set of exogenously given constraints imposed on the decision-maker?

Just to be a bit clearer on what we mean by abstract properties, consider a budget constraint. It represents what the decision-maker can afford (i.e., what the decision-maker can purchase and, hence, what the decision-maker can obtain without the use of illegal methods such as stealing). Now a budget contraint does have some abstract properties. For example, if a new good is introduced in the market then, all else equal, what the decision could afford previously the decision-maker can still afford after the good is introduced. This follows because if the decision-maker does not buy any of the newly introduced goods, then the decision-maker has not spent any of his wealth on these goods. Provided that prices and his wealth remain the same, he can afford any bundle of goods that he could afford before. This is a typical example of a principle that underlies, only in part naturally, what we refer to as a theory of the consumer. What we are looking for are equally simple and obvious abstract principles that could underlie a broader (axiomatized) theory of decision-making under the law.

Consider the decision function D of a law abiding citizen (i.e., D(B)= argmax u(x) subject to x ∈ L(B)). So, this is the decision function of an agent who abides by the law L and subject to that constraint, maximizes a utility function. In order to axiomatize the choices of a law abiding citizen, it is necessary to determine what are the properties, if any, that one can assume for the law L. A stumbling block in this endeavor is that no matter how simple and intuitive is the abstract principle that we have considered, we could always show, at least so far, that the law violates it. For example, consider the idea that if the law deems an option to be illegal to a decision-maker, then there must be something else that the decision-maker could have done that is deemed to be better (in the traditional sense of better and worse given by an asymmetric binary relation ≻). This may seem a natural candidate for an abstract principle of the law. Yet, as we show, the law violates it.

Now clearly, we cannot show a negative. That is, we cannot show that the law violates any abstract principle. Nor do we believe in such conclusion. It seems (to us) quite implausible that the law does not satisfy any principle. It seems far more plausible that we simply have not found these general principles yet. It would also be quite tedious to describe to the reader all principles that we considered so far, but discarded. Hence, our fourth and last example will just illustrate the type of principle we are looking for, but ultimately discarded because the law does not satisfy it. Consider three options x, y and z. If all options are legal in all binary choices (i.e., two options are feasible), then any of three options are legal when they are all feasible. We refer to this principle as the indifference axiom. It is weaker than the axioms we typically find in the consideration sets of bounded rationality theories in economics and the law violates it."

In the introduction, we also added some lines to motivate some of the specific properties that we address. They read "The first example shows that the law violates WARP. Thus, this example cast doubt on the often repeated idea that the law maximizes a social welfare function. The second example shows a deeper violation of WARP: the law does not maximize an asymmetric binary relation. Hence, the law violates the seemingly natural principle that "illegal options are the options deemed inferior to other feasible options," if by inferior we mean that for some asymmetric binary relation ≻, x ≻y whenever an option y is deemed inferior to an alternative option x."

Finally, we added section 5 in this revision. In section 5 we do axiomatize a model of legal choices, when the law is just an exogenously given constraint. But we conclude that exercise as follows: "Remark 3 shows how an axiomatized theory of law-abiding choices when the consideration set of legal options is arbitrary and is not assumed to have any underlying abstract properties. As it is easy to see, this is not an interesting theory of law-abiding choices in the same way that it would not be an interesting choice of limited attention if we assume limited attention to be just any exogenously given constraint. The development of a proper theory of law abiding choices required the determination of some general properties of the law, if they exist. However, as our examples show, it is not an easy task to determine what these properties might be."

In your report, you also mentioned some typos in our paper. We are grateful to you for finding them. Naturally, we fixed them in this revision. For completeness, we now go over each of them (numbered) and how we revised the paper accordingly.

A short list of typos.
1) (line 1, page 6) Remove the dot after "et" and the comma after "al." in the reference.

Now Fixed!

2) (line 9, page 6) Is the statement in boldface an other example? Why does it have a proof while the others do not.

Our use of boldface were a bit inconsistent in the previous version of the paper. In this revision, we differentiate examples and remarks more distinctly. Naturally, both examples and remarks have proofs. It is just that remarks do not involve new examples (as in remark 1) and sometimes the proofs of the remarks are a bit more involved and so we ended them with a marker ■ to show when the proofs end.

3) (definition 3, page 7) Write "three" instead of "thre".

Now Fixed!

4) (line 4, Section 5, page 8) Write "different" instead of "differ". Or remove "are".

Now Fixed!

5) (line 1, Appendix, page 8) Is the statement in boldface an other example? Why does it have a proof while the others do not.

In this revision, we removed the boldface for both examples and remarks, and we provide a short proof for remarks and examples, as they are needed.

6) One more comment about presentation.
Why do you need such a short appendix? Why can't that text be moved to the main part?

In this revision, we moved the proofs to the main text.

We are grateful to you for the detailed comments that you made on our work. This revision is much improved because of your comments and suggestions. We hope that we revised the paper in the lines that you expected and that we took full advantage of your report.

Round 2

Reviewer 2 Report

I appreciate the author's detailed response and the changes that s/he has introduced into the manuscript. I am still not convinced why anyone would expect "the law" (i.e, any real law) to have the listed properties, but I consider that my job, as a referee, is to express my opinion on the scientific soundness of the analysis. So, I am going to let the editor judge the relevance of the underlying research questions.

Let me however explain 1) why I am not convinced by the presented argument and 2) also point out some remaining formatting issues.

1)
The author claims that s/he considers the law as an abstract construct. Yet at the bottom of page 6, it says that "We describe the las, as the law is (Model Penal Code, 3.04 and 3.06)." That does look like a very specific example to work with. How about the law in Nigeria, Japan, or Finland? 

Let me also comment on the misleading comparison of the law to the budget constraint. The legal system is introduced/defined as a mapping (L) in the paper. The author then claims that "[w]e may refer to L as the law." The confusion arises as the author shows that the law in real life does not have all the nice properties that the mapping L could (and maybe should) have in theory.
Well, isn't that the price we always have to pay when working with abstract models? They are always wrong, by their very nature. Yet, we might be able to derive interesting conclusions from  them. Even if it appears as a neat straight line in the two-dimensional textbook examples, the budget constraint can, for example, also be "soft" in many real-life settings. All we need is to make sure that we use the correct model for the problem that we are trying to solve. Maybe certain laws in certain countries do have those esteemed properties.

2)
(page 1, line 7) There is too much empty space after "a" at the beginning of the line.
(page 1, paragraph 2) I find it confusing that the text between parentheses includes part of a sentence and two separate full sentences.
(several locations) There is too much space after "et al.".
(page 5, third line after Definition 2) The dot at the end of the sentence seems to appear in boldface.
(Remarks) The text in the second line of the Remarks should start at the left margin, just like in the Definitions.
(page 8) Why is there a black box after the first paragraph?
(page 9) Why is there a black box after the first paragraph below Remark 2?
(page 11) Why does Remark 3 have a "Proof"? Remark 1 and 2 do not have one.

Author Response

Dear Referee:

Thank you for the detailed and constructive report on our paper. We have revised the paper according to your guidelines and we believe that this revision is now much improved. We now reproduce your report, followed by the line "How we revised our paper according to your guideline.''

I appreciate the author's detailed response and the changes that s/he has introduced into the manuscript. I am still not convinced why anyone would expect "the law" (i.e, any real law) to have the listed properties, but I consider that my job, as a referee, is to express my opinion on the scientific soundness of the analysis. So, I am going to let the editor judge the relevance of the underlying research questions.

Let me however explain 1) why I am not convinced by the presented argument and 2) also point out some remaining formatting issues.

1)
The author claims that s/he considers the law as an abstract construct. Yet at the bottom of page 6, it says that "We describe the las, as the law is (Model Penal Code, 3.04 and 3.06)." That does look like a very specific example to work with. How about the law in Nigeria, Japan, or Finland? 

"How we revised our paper according to your guideline.''  Your concern that there might be several types of law is quite significant. In page 4, second last paragraph of this revision, we now add: ``Some readers may be concerned with the idea of the law in the singular. The law can differ in different times and places. However, our examples are based on widely shared legal principles. After each example, we discuss the generality of these basic principles." In particular, our examples work in the law in Nigeria, Japan, and Finland. Nigeria uses the British Penal Code and Japan uses the German code. We do not know about Finland, but we are certain that our examples apply to Finland as well. The law typically divides itself in two categories: Malum in se (wrong in itself) and Malum Prohibitum (wrong only because it is prohibited). In the second category, there is indeed great variation in the laws of different countries and time periods, but malum in se refers to basic restrictions such as the condemnation of murder which are far more general and universal. There are not that many variations there. All our examples deal with the most basic doctrines of the law and they apply with great generality. To be sure, there can be some limitations to our examples. Therefore, in this revision, we added after each example, a part of the paper entitled "the legal principles in example (1,2,3).'' There we refer to the basic legal doctrines that we use to generate each example and the few limitations that our examples might have. ...................................................................................................................... Let me also comment on the misleading comparison of the law to the budget constraint. The legal system is introduced/defined as a mapping (L) in the paper. The author then claims that "[w]e may refer to L as the law." The confusion arises as the author shows that the law in real life does not have all the nice properties that the mapping L could (and maybe should) have in theory. Well, isn't that the price we always have to pay when working with abstract models? They are always wrong, by their very nature. Yet, we might be able to derive interesting conclusions from  them. Even if it appears as a neat straight line in the two-dimensional textbook examples, the budget constraint can, for example, also be "soft" in many real-life settings. All we need is to make sure that we use the correct model for the problem that we are trying to solve. Maybe certain laws in certain countries do have those esteemed properties. "How we revised our paper according to your guideline.''  It is true that models are incorrect, but that does not mean that any assumption is valid. A map is not a perfect representation of a terrain, but it should not point east to what is actually west. Our examples are not based on eccentric legal doctrines or some exceptions to the law. On the contrary, they are based on basic legal doctrines that are shared by a wide variety of cultures and are rooted on deep moral principles. Hence, to address our concern, we eliminated example 4 from this paper (the example that showed that the law does not satisfy the indifference axiom). That example seemed (to us) a bit orthogonal to the main point of the paper and to models of bounded rationality such as limited attention. Finally, that example is based on many legal doctrines and so it is as clearly rooted in elementary doctrines as in the first three examples ...........................................................................................................................................................................

2)
(page 1, line 7) There is too much empty space after "a" at the beginning of the line.
(page 1, paragraph 2) I find it confusing that the text between parentheses includes part of a sentence and two separate full sentences.
(several locations) There is too much space after "et al.".
(page 5, third line after Definition 2) The dot at the end of the sentence seems to appear in boldface.
(Remarks) The text in the second line of the Remarks should start at the left margin, just like in the Definitions.
(page 8) Why is there a black box after the first paragraph?
(page 9) Why is there a black box after the first paragraph below Remark 2?
(page 11) Why does Remark 3 have a "Proof"? Remark 1 and 2 do not have one.

"How we revised our paper according to your guideline.''  Thank you for pointing these errors to us. We corrected them in this revision. The only exception was "too much space after "et al.". We could not see the issue, but we hope that it is fine in this revision. A few comments: The black box was an indication of the end of the proof. Perhaps this is obvious and so we just removed it. Remark 3 is an easy proposition. This why it had a proof and we called it a remark. It is now called a proposition in this revision.   We would like to thank you, once again, for the careful and constructive report on our paper. We hope that this revision is in the lines that you expected and that we took full advantage of our report. We believe that the manuscript is now much improved, and we would like to thank you, once more, for your help and guidance in this revision.   Best regards,   Alvaro and Leo.